# Long Non-Coding H19 in Lymphocytes: Prognostic Value in Acute Ischemic Stroke Patients

**DOI:** 10.3390/ph17081008

**Published:** 2024-07-31

**Authors:** Liyuan Zhong, Zixian Xie, Ziping Han, Junfen Fan, Rongliang Wang, Zhen Tao, Qingfeng Ma, Yumin Luo

**Affiliations:** 1Institute of Cerebrovascular Disease Research, Xuanwu Hospital of Capital Medical University, Beijing 100053, China; zhongliyuan@ccmu.edu.cn (L.Z.); xiezx@mail.ccmu.edu.cn (Z.X.); hanzip@xwhosp.org (Z.H.); fanjunfen@xwhosp.org (J.F.); wangrongliang@xwhosp.org (R.W.); taozhen@xwhosp.org (Z.T.); 2Department of Neurology, Xuanwu Hospital of Capital Medical University, Beijing 100053, China; 3Beijing Geriatric Medical Research Center, Beijing Key Laboratory of Translational Medicine for Cerebrovascular Diseases, Beijing 100053, China

**Keywords:** AIS, lncRNA H19, lymphocytes, prognosis

## Abstract

Acute ischemic stroke (AIS) is a cerebrovascular disease that seriously affects the physical and mental health and quality of life of patients. However, there is a lack of reliable prognostic prediction methods. The main objective of this study was to investigate the prognostic value of long non-coding RNA (lncRNA) H19 in lymphocytes of patients with AIS, and to construct a prognostic prediction model for AIS including lncRNA H19 in lymphocytes, which would provide new ideas for the prognostic evaluation of AIS. Poor prognosis was defined when the patient’s modified Rankin scale (mRS) score at 3 months after AIS onset was greater than 2. Quantitative real-time polymerase chain reaction (qRT-PCR) was used to measure the level of lncRNA H19 in lymphocytes. Spearman correlation analysis revealed a positive correlation between lncRNA H19 and mRS score at 3 months after AIS onset (r = 0.1977, *p* = 0.0032), while lncRNA H19 was negatively correlated with white blood cells counts, lymphocytes counts, and neutrophils counts. Logistic regression analysis identified lncRNA H19 as an independent predictor of poor prognosis (OR = 3.062 [1.69–5.548], *p* < 0.001). Moreover, a nomogram prediction model incorporating lncRNA H19 in lymphocytes demonstrated effective discrimination, calibration, and clinical applicability in predicting AIS outcomes. The findings suggest that lncRNA H19 in lymphocytes could be a valuable prognostic indicator and a potential pharmacological target for AIS patients, and might be a novel pathway for enhanced prognostic evaluation and targeted therapeutic strategies.

## 1. Introduction

Acute ischemic stroke (AIS), triggered by systemic hypoperfusion, local thrombosis, or embolism, is recognized as the most common type of cerebrovascular disorder [1]. Known for its high mortality rate, AIS presents a serious danger to the survival of patients. In addition to posing an immediate threat to life, AIS also significantly impacts the physical and psychological well-being of patients, leading to a marked decline in their quality of life. It has emerged as a critical global public health issue, resulting in enormous social and economic burdens. According to a recent report, the number of stroke deaths worldwide is expected to increase by 50 percent to 9.7 million a year by 2050, resulting in economic losses of up to $2.3 trillion [2]. Thus, accurately predicting the prognosis of patients with AIS is of paramount importance to both the patients and medical personnel. However, there is no reliable prognostic method for assessing the poor prognosis of AIS. Therefore, it is imperative to discover reliable and accurate biomarkers for predicting the prognosis of acute ischemic stroke, to enhance the precision of prognostic risk assessment, and to assist clinicians in identifying pharmacological targets and devising personalized treatment plans, as well as to optimize the overall prognosis and improve health outcomes for patients with AIS.

Long non-coding RNAs (lncRNAs) represent a category of non-coding RNAs surpassing 200 nucleotides in length [3]. Dharap and colleagues identified the abnormal expression of numerous lncRNAs within the cerebral cortex of rats following cerebral ischemia by using microarray chips [4]. On this basis, the Dykstra–Aiello research team has documented the irregular expression profiles of a range of lncRNAs in the peripheral blood samples of individuals suffering from ischemic stroke [5]. These findings underscore the potential of lncRNAs as a promising class of biomarkers and therapeutic targets, significantly enhancing the clinical prediction of outcomes for patients with AIS. LncRNA H19 stands out as one of the earliest recognized imprinted genes, renowned for its high degree of conservation [6]. In a study conducted in 2015, researchers employed post-mortem tissues samples from five donors to examine the expression patterns and stability of lncRNAs across three distinct brain regions: the cortex, white matter, and cerebellum. Their findings reveal that lncRNA H19 is one of the top ten most stably expressed lncRNAs in the human brain [7]. A recent study has unveiled a compelling link between the genetic variation of the lncRNA H19 and the occurrence of ischemic stroke [8]. It is particularly noteworthy that the blood level of lncRNA H19 was significantly higher in patients with acute ischemic stroke compared to healthy subjects [9]. Furthermore, foundational studies have illuminated the critical role of lncRNA H19 in the intricate pathophysiological processes associated with ischemic stroke. It influences pivotal mechanisms, including inflammation, oxidative stress, apoptosis, autophagy and neurogenesis [10,11,12,13,14], which in turn highlights its capacity to function as a reliable prognostic biomarker and a viable pharmacological target for AIS, underscoring its significant role in advancing clinical management and offering insights into patient outcomes and guiding therapeutic strategies. Studies have found that an upregulated H19 level in plasma and exosomes was associated with poor prognosis in patients with ischemic stroke [15,16]. Nevertheless, the RNA level in blood plasma is considerably lower than those found in peripheral blood cells, and it is susceptible to the influence of anticoagulant substances [17]. Lymphocytes were reported to infiltrate the central nervous system (CNS) in experimental stroke, and can be detected over a period of up to 12 weeks [18]. A multitude of clinical studies have also underscored the tight correlation between the neutrophil-to-lymphocyte ratio (NLR) and the clinical outcomes of AIS patients [19,20,21]. These findings indicate that H19 expression in lymphocytes might be closely related to long-term prognosis after AIS. Nevertheless, to date, no research has documented the link between H19 expression in lymphocytes and the prognosis of AIS.

Therefore, this study is designed to explore the prognostic significance of lncRNA H19 within lymphocytes in AIS patients. Key objectives of our study include the development and validation of an AIS prognostic prediction model that incorporates lncRNA H19 level in lymphocytes, along with the provision of a personalized prognosis calculator for individual patients. The ultimate objective is to equip healthcare professionals with a more refined and tailored approach to evaluate the prognostic risks faced by AIS patients. This will allow for the prompt recognition of those at higher risk of poor prognosis, enabling timely and precise pharmacological interventions to enhance their chances of a good prognosis.

## 2. Materials and Methods

### 2.1. Study Participants

Ethical clearance for this study was granted by the Ethics Committee of Xuanwu Hospital, Capital Medical University. Our research population comprised patients who were diagnosed with AIS by experienced neurologists at Xuanwu Hospital, Capital Medical University, between November 2018 and September 2019. The inclusion criteria for the study were as follows: 1. Participants aged 18 to 99 years at the onset of symptoms; 2. Diagnosis of AIS by neurologists based on a combination of clinical presentations and diagnostic imaging results, including cranial computed tomography or magnetic resonance imaging; 3. The time of symptom onset was within a 24 h window. Exclusion criteria included: 1. Absence of admission NIH Stroke Scale (NIHSS) scores; 2. A diagnosis of cancer, infectious disease, or autoimmune disorder within the four weeks preceding study enrollment; 3. Loss to follow-up during the first three months post-admission period; 4. Degradation of RNA in the collected samples. In total, 221 AIS patients were enrolled in this study. Informed consent was obtained from all patients or their legal representatives.

### 2.2. The Collection of Clinical Data

Baseline demographic and clinical data, including age at AIS onset, gender, time of stroke onset, systolic and diastolic blood pressures, and a comprehensive review of medical histories such as hypertension, diabetes, coronary heart disease, and atrial fibrillation, were extracted from the hospital’s electronic medical records. Additionally, a series of laboratory parameters were included, including white blood cell counts, neutrophil counts, lymphocyte counts, platelet counts, and blood lipid profiles. The stroke severity was rated by well-trained clinicians using the NIHSS during the initial 24 h post-admission [22].

At the 90-day mark following the AIS event’s onset, patient prognoses were ascertained via telephone consultations with either the patients or their family members. The functional statuses of AIS patients were appraised by expert neurologists using the modified Rankin Scale (mRS) [23], and these experts were kept unaware of the patients’ medical backgrounds. This assessment was based on narratives provided by the patients and their relatives. An mRS score of 3 to 6 indicated a poor prognosis, whereas a score of 0 to 2 denoted a good one.

Based on the trial of ORG 10172 on the Acute Stroke Treatment (TOAST) classification system, AIS cases were categorized into five distinct subtypes: large artery atherosclerotic stroke, cardioembolic stroke, small vessel occlusion, stroke of other determined etiology, and stroke with undetermined etiology [24].

### 2.3. RNA Extraction and Quality Detection

Venous blood from patients with AIS was collected by an experienced nurse into an anticoagulation tube containing ethylenediaminetetraacetic acid (EDTA) before starting treatment. The blood was carefully dripped down the side of the pre-filled centrifuge tube with human lymphocytes separation medium (Haoyang Biotech, Tianjin, China). The tube, now containing the mixture, was then placed into a centrifuge and spun at 4 °C and 3000 rpm for 10 min. After centrifugation, the liquid exhibited a distinct layering phenomenon. The cloudy cell layer in the middle of the centrifuge tube was taken out with a pipette and put into a new centrifuge tube containing red blood cell lysis buffer. The tube was put on ice for 15 min and mixed a few times with a vortex mixer. Subsequently, the sample tubes were centrifuged at 4 °C and 3000 rpm for 10 min to collect the lower sediment layer, representing the lymphocytes. The lymphocytes were then washed with 1 mL of physiological saline and transferred to a new 1 mL RNase-free EP tube. After another centrifugation at 4 °C and 3000 rpm for 10 min, the supernatant was discarded. Next, 1 mL of TRIzol reagent was added to each centrifuge tube, and the samples were preserved at −80 °C until use.

The samples were retrieved from the −80 °C freezer and placed on ice to thaw. Subsequently, 200 µL of chloroform was added to each EP tube, followed by vigorous shaking for 15 s and 15 min of incubation at 25 °C until phase separation occurred. The EP tubes were then centrifuged at 4 °C and 12,000 rpm for 15 min. After centrifugation, the supernatant was carefully transferred to a new sterile, RNase-free EP tube, and 500 µL of isopropanol was added. The mixture was then incubated at −20 °C overnight. The following day, the EP tubes were centrifuged again at 4 °C and 12,000 rpm for 10 min. After centrifugation, the supernatant was discarded, and each EP tube was washed with 1 mL of 75% ethanol, a process repeated twice, followed by the removal of the ethanol solution. The EP tubes were then placed in a clean bench and air-dried at 25 °C for 30 min. Subsequently, 40 µL of DEPC-treated water was added to each EP tube, and the tubes were incubated in a 55 °C water bath to facilitate RNA dissolution. The purity and concentration of the extracted RNA were assessed using a spectrophotometer, measuring the absorbance ratios at 260/280 and 230/260. The integrity of the RNA was evaluated by agarose gel electrophoresis to determine if any degradation had occurred.

### 2.4. cDNA Synthesis and Real-Time Polymerase Chain Reaction (RT-PCR)

The cDNA was synthesized from total RNA using a SuperScript III Reverse Transcriptase Kit (Invitrogen, Carlsbad, CA, USA). Subsequently, quantitative RT-PCR was conducted on the QuantStudio 5 Real-Time PCR System (Applied Biosystems, Waltham, MA, USA) using the SYBR Green PCR Master Mix (YEASEN, Shanghai, China). The amplification of the housekeeping gene β-actin was used as an endogenous control. The RT-PCR primers for human β-actin were Forward (F) 5′-GTGGCCGAGGACTTTGATTG-3′ and Reverse (R) 5′-CCTGTAACAACGCATCTCATATT-3′. The RT-PCR primers for human H19 were Forward (F) 5′-ACGTGACAAGCAGGACATGACA-3′ and Reverse (R) 5′-ACCAGCCTAAGGTGTTCAGGAA-3′. The PCR reaction was performed under the following cycling conditions: 95 °C for 10 s, 60 °C for 60 s, with a final step of 95 °C for 15 s, followed by a gradual increase from 60 °C to 99 °C. For each assay run, standard curves were constructed using a dilution series of the control cDNA. The level of gene expression was determined using the data from the standard curve. Each sample was tested three times and the average Ct-value was documented. The data were evaluated using the comparative threshold cycle method (2^−ΔΔCT^) [25], with normalization to the endogenous reference gene β-actin, and presented as relative values (ΔΔ CT) in comparison to the control sample.

### 2.5. Statistical Analysis

These 221 individuals were categorized into good prognosis and poor prognosis groups. The baseline characteristics and the expression levels of lncRNA H19 within lymphocytes were compared between the two groups. For the two groups that were normally distributed, an independent samples t-test was used for comparison; otherwise, the Mann–Whitney U test was applied. Categorical variables were compared using the Chi-square test or Fisher’s exact test. Correlations between variables were analyzed using the Spearman rank correlation coefficient. To identify predictors significantly associated with poor prognosis in patients with AIS, both univariate and multivariate logistic regression analyses were employed. Variables with a *p*-value < 0.05 in the univariate logistic analysis were included in the multivariate logistic regression, using a backward stepwise regression method for exploring predictive factors. A nomogram model for predicting the risk of poor prognosis in patients with AIS was constructed based on the independent factors selected by multivariate logistic regression, and the model’s discriminative ability, calibration performance, and clinical utility were evaluated using the receiver operating characteristic (ROC) curve, calibration curve, and decision curve analysis (DCA).

## 3. Results

### 3.1. Baseline Characteristics of Patients Grouped by 3-Month Functional Outcomes

After applying the inclusion and exclusion criteria, a total of 221 patients with AIS were enrolled in the study. These individuals were stratified into good and poor prognosis groups based on their mRS scores at three months post-admission, with 138 individuals in the good prognosis group and 83 in the poor prognosis group. Subsequently, a comparative analysis was conducted on their baseline characteristics and the expression level of lncRNA H19 in lymphocytes. The baseline characteristics between the two groups are presented in Table 1. The median age of the good prognosis group was lower than the median age of the poor prognosis group (62.50 [54.75–71.00] vs. 70.00 [60.00–82.00], *p* < 0.001 *). The median admission NIHSS scores were lower in the good prognosis group than those in the poor prognosis group. (4.00 [2.00–7.00] vs. 12.00 [7.00–17.00], *p* < 0.001 *). The probability of receiving endovascular mechanical thrombectomy (EMT) was lower in the good prognosis group than in the poor prognosis group (10.87% vs. 22.89%, *p* = 0.016 *). The prevalence of coronary heart disease (15.22% vs. 32.53%, *p* = 0.003 *) and atrial fibrillation (11.59% vs. 27.71%, *p* = 0.002 *) was lower in the good prognosis group than in the poor prognosis group. Furthermore, the lymphocytes count was higher (1.73 [1.25–2.22] vs. 1.31 [0.91–1.90], *p* < 0.001 *), while the neutrophils count (4.72 [3.70–5.93] vs. 5.95 [4.04–7.56], *p* = 0.006 *) and the neutrophil-to-lymphocyte ratio (NLR) (2.61 [1.77–4.33] vs. 4.14 [2.34–7.43], *p* < 0.001 *) was lower, in the good prognosis group than that in the poor prognosis group. In terms of the expression level of lncRNA H19 in lymphocytes, the expression level in the good prognosis group was lower than that in the poor prognosis group (0.69 [0.51–1.07] vs. 1.19 [0.66–1.54], *p* < 0.001 *).

### 3.2. Relationship between the H19 Expression Level and Counts of Circulating Cells

In order to gain a deeper understanding of the relationship between H19 expression level in lymphocytes and circulating cell populations, a correlation analysis was conducted using Spearman correlation analysis to evaluate the statistical significance of the association between H19 level in lymphocytes and these cell populations. The results indicate that the expression level of H19 in lymphocytes was negatively correlated with the counts of white blood cells (r = −0.1946, *p* = 0.0037, Figure 1A), neutrophils (r = −0.1564, *p* = 0.0200, Figure 1B), and lymphocytes (r = −0.1501, *p* = 0.0256, Figure 1C).

### 3.3. H19 Level in Lymphocytes Was Decreased in AIS Patients and Associated with Prognosis of AIS Patients

The correlation between the level of H19 in lymphocytes and the mRS scores at three months post-admission was evaluated using the Spearman correlation analysis. The results demonstrate a positive correlation between the level of H19 in lymphocytes and the mRS scores at three months (r = 0.1977, *p* = 0.0032, Figure 2A), suggesting that H19 in lymphocytes was associated with clinical prognosis. Patients with poor prognosis (mRS > 2) exhibited significantly higher levels of H19 in lymphocytes compared to those with good prognosis (mRS ≤ 2) (*p* < 0.001, Figure 2B).

### 3.4. H19 Used as an Independent Predictor for Poor Prognosis in AIS

The results of the univariate logistic regression analysis indicate that age, admission NIHSS scores, EMT, coronary heart disease, atrial fibrillation, neutrophils counts, lymphocytes counts, NLR, and the level of lncRNA H19 in lymphocytes were all identified as independent risk factors associated with poor prognosis in patients with AIS (Table 2).

The results of multivariate logistic regression analysis reveal that age (OR = 1.042 [1.012–1.073], *p* = 0.006), admission NIHSS scores (OR = 1.334 [1.211–1.470], *p* < 0.001), EMT (OR = 0.342 [0.112–1.046], *p* = 0.060), NLR (OR = 1.102 [0.998–1.217], *p* = 0.054), and lncRNA H19 level in lymphocytes (OR = 3.062 [1.690–5.548], *p* < 0.001) were identified as independent predictors of poor prognosis in AIS patients (Figure 3).

### 3.5. Incremental Predictive Value of lncRNA H19 in Lymphoctes for Poor Prognosis of AIS

Furthermore, the incremental predictive value of lncRNA H19 in lymphocytes for poor prognosis in AIS was assessed using the area under the curve (AUC), integrated discrimination improvement (IDI), and net reclassification index (NRI). The results demonstrate that incorporating lncRNA H19 expression level in lymphocytes into the multifactorial regression model resulted in a significant increase in AUC, from 87% to 89% (Table 3). In addition, the inclusion of lncRNA H19 within the clinical model resulted in a significant improvement in the overall IDI value (5.03% [1.76–8.30%], *p* = 0.0026) and the continuous NRI (68.33% [42.86–93.79%], *p* < 0.0001). Moreover, the new risk prediction model demonstrated enhanced model fit, as evidenced by its superior R-squared (Cox and Snell) value of 0.409, surpassing the traditional model’s R-squared (Cox and Snell) of 0.366. Specific outcomes of the multiple logistic regression analyses and an evaluation of the predictive capabilities between the two models have been presented in Table 3.

### 3.6. Creation and Evaluation of a Predictive Tool Based on the Independent Prognostic Factors

In order to facilitate and enhance the precision of predicting the individualized clinical prognosis for AIS patients, a nomogram was developed (Figure 4A). This nomogram was based on the five independent prognostic factors in a multifactorial logistic regression model. The utilization of the nomogram was straightforward and could be executed as follows: (1) align a value for each variable along its respective axis with the Points axis at the top to identify the corresponding point total for that variable’s specific position on the nomogram; (2) add the scores attributed to each covariate and pinpoint the resultant total on the Total Points axis to derive a total score; (3) extend a perpendicular line from the Total Points axis to the Risk axis to delineate the individual’s risk of poor prognosis.

Subsequently, the discriminative ability, calibration performance and clinical utility of the aforementioned nomogram model were evaluated through the use of the ROC curve, calibration curves and DCA. This prognostic nomogram demonstrated significant discriminative abilities, as evidenced by an AUC of 0.890 (Figure 4B). Additionally, a calibration curve was constructed that places the predicted risk along the x-axis and the observed risk along the y-axis (Figure 4C). This graphical representation shows the strong agreement between the model’s prognostications and empirical data. To further assess the clinical efficacy of the nomogram model, a DCA was executed. The analysis depicts that the net benefit on the vertical axis corresponds to the probability threshold for poor prognosis prediction on the horizontal axis. The DCA outcome reveals that, for a wide spectrum of threshold probabilities, the nomogram model outperformed the universal “Treat All” or the null “Treat None” approaches, thereby underscoring its clinical superiority (Figure 4D).

## 4. Discussion

In this study, we examined the expression levels of the lncRNA H19 in the lymphocytes of 221 AIS patients, and explored its association with prognosis. The genomic structure of the human lncRNA H19 is located within the 1995 Kbp to 2000 Kbp region on chromosome 11, at the GRCh38.p12 primary assembly [26]. H19, which encompasses five exons and four introns, is one of the earliest identified imprinted genes with a high degree of conservation [27]. A previous study indicated that lncRNA H19 is among the top ten most stably expressed lncRNAs in the human brain [7]. Initially, we employed the Spearman correlation method to analyze the relationship between the lncRNA H19 level in lymphocytes and the counts of white blood cells, neutrophils, and lymphocytes in the circulatory system. The results demonstrate a negative correlation between the expression level of H19 and the counts of white blood cells, neutrophils, and lymphocytes within the circulation. Subsequently, we investigated the relationship between the expression level of H19 in lymphocytes of AIS patients and their prognosis. Using the Spearman method, we analyzed the correlation between the H19 expression level and the mRS scores at 3 months post-admission. The findings reveal a positive correlation between H19 expression level in lymphocytes and the mRS scores at 3 months post-admission, strongly suggesting a close relationship between the expression level of H19 in lymphocytes and the prognosis of AIS patients. Previous studies have also reported findings that are consistent with our results. Wang et al. found that a high level of plasma exosomal H19 was related to high mRS scores 7 days after AIS onset [15]. Lapikova-Bryhinska et al. revealed that a high level of lncRNA H19 in plasma was associated with a higher death risk in patients with AIS [16]. However, the amount of RNA in the blood plasma is significantly lower than in peripheral blood cells, and it can be easily affected by anticoagulant agents [17]. According to previous reports in the literature, lymphocytes infiltrate the CNS in a delayed manner in experimental stroke and can be observed for as long as 12 weeks [18]. Therefore, we believe that lncRNA H19 in lymphocytes is more suitable for use as a biomarker for predicting the long-term functional prognosis of AIS patients than that in plasma or exosomes.

Subsequently, we intend to stratify patients into two groups based on favorable and unfavorable prognosis and compare the differences in H19 level in lymphocytes between these two groups. A previous study has shown that when patients move from a state of severe disability status of mRS 4–5 to a mild disability status of mRS 2, their quality of life and self-sufficiency enhance significantly [28]. Such an improvement is immensely beneficial for the individuals and society as a whole. Additionally, this transformation signifies that patients are empowered to take better care of themselves, regain autonomy in their daily routines, and significantly improve their overall quality of life. At the same time, this improvement reduces the demand for long-term care and support, thereby alleviating the burden on families and society. Furthermore, this shift also facilitates better social integration for patients, improving their social and work capabilities, leading to greater life satisfaction and happiness. In summary, the transition from severe to mild disability is not only of great significance to the individual patient but also has a positive impact on society as a whole. Therefore, we define an mRS score of 3–6 at 3 months post-admission as “poor prognosis” after acute ischemic stroke, and an mRS score of 0–2 as “good prognosis”. The 221 participants were categorized into two groups according to their 3-months-post-admission results. We compared the H19 expression levels in lymphocytes between the two groups using the Mann–Whitney U test. Our findings reveal that patients with a poor prognosis displayed elevated H19 expression in lymphocytes compared to those with a good prognosis, indicating a strong association between high H19 expression and poor prognosis. These findings highlight the potential significance of lncRNA H19 in the context of functional recovery following a stroke. These findings are consistent with those from our previous basic research on H19 in the context of AIS. Wang et al. found that lncRNA H19 suppresses Notch 1 expression by inhibiting p53’s transcriptional activity, thereby curbing neurogenesis in the subventricular (SVZ) and subgranular zones (SGZ) post-ischemic stroke [14]. Additionally, a previous study demonstrated that the inhibition of lncRNA H19 has been observed to augment cell viability and attenuate the number of cells undergoing apoptosis in the SH-SY5Y cell model subjected to oxygen–glucose deprivation and reperfusion (OGD/R) [29]. Previous studies found that lncRNA H19 regulated the secretion of inflammatory factors, such as TNF-α, TGFβ1, IL-2, IL-4, IL-6 and IL-10 [30,31,32]. Interleukins such as TNF-α, IL-2, IL-4, IL-6 and IL-10, secreted by lymphocytes, play an important role in the pathogenesis and prognosis of ischemic stroke [33,34]. Hence, we speculated that H19 expression specifically in lymphocytes might regulate inflammation by controlling the level of inflammatory factors secreted by lymphocytes in AIS.

Subsequently, we employed univariate and multivariate logistic regression analyses to identify potential risk factors associated with poor prognosis in AIS patients. Univariate analysis revealed that admission NIHSS scores, age, EMT, coronary heart disease, atrial fibrillation, neutrophils counts, lymphocytes counts, NLR, and the expression level of H19 in lymphocytes were correlated with poor prognosis in AIS patients. Then, variables with *p*-values less than 0.05 according to univariate analysis were included in a multivariate analysis using a backward stepwise regression method to select independent risk factors associated with poor prognosis in AIS patients. Previous studies have indicated that factors such as admission NIHSS scores [35,36], age [37,38], EMT [39,40], and NLR [19,21] are risk factors for poor prognosis in AIS. In alignment with these findings, our study has also demonstrated that age, admission NIHSS scores, EMT, NLR, and the level of H19 in lymphocytes were independently associated with poor prognosis in AIS. Moreover, our study particularly highlighted the independent correlation between the expression level of H19 in lymphocytes and poor prognosis in AIS. The incorporation of H19 into the clinical model significantly enhanced the reclassification capacity, with improvements in AUC, continuous NRI, and IDI. The findings of this study suggest that the integration of lncRNA H19 within lymphocytes into prognostic models for AIS patients offers a significant improvement in predictive accuracy. An enhanced AUC, along with positive IDI and NRI values, indicated that lncRNA H19 is a valuable biomarker for the stratification of AIS patients’ risk of poor prognosis. An increased R^2^ (Cox and Snell) value further supported the significance of lncRNA H19 in predicting AIS prognosis. These results suggest that lncRNA H19 may serve as a valuable prognostic predictor for the three-month functional outcome in AIS patients.

Based on the independent predictive factors identified in the aforementioned multivariate logistic regression model, we proceeded to construct and validate a nomogram model for predicting the three-month neurological functional prognosis of AIS patients. This nomogram prediction model can potentially assist neurologists in providing a more precise personalized assessment of the risk of prognosis for AIS patients. We subsequently evaluated the predictive performance of the nomogram through ROC analysis, calibration curves, and DCA. The validation results emphasize the nomogram’s strong discriminative ability, clinical utility, and predictive accuracy. The model’s high discriminative power, calibration accuracy, and clinical impact highlight its potential utility as a reliable tool for personalized prognostic prediction in clinical practice.

However, we must acknowledge that our study does have certain limitations. To begin with, the study was confined to a single hospital setting, which could potentially limit the broader applicability of our conclusions. Moreover, the small number of AIS included indicates a need for caution when generalizing these findings to a wider audience. Additionally, despite our diligent efforts to account for a variety of potential confounding elements, it is possible that some unconsidered factors may have had an impact on our results.

Nevertheless, we maintain that the findings of our study offer substantial value to neurologists and AIS patients themselves. The findings of our study empower neurologists to administer a more effective, personalized evaluation of prognostic risks for AIS patients. This could lead to the timely recognition of those at an elevated risk of poor prognosis, allowing for more targeted and proactive care. Ultimately, these efforts could contribute to a more favorable prognosis for AIS patients.

## 5. Conclusions

In summary, our research underscores the potential of lncRNA H19 as a novel and promising biomarker, offering a new avenue for both prognostic assessment and as a potential pharmacological target in the context of AIS. Furthermore, the innovative nomogram model developed in our study provides a more comprehensive and personalized approach to the evaluation and management of AIS patients, enhancing clinical decision-making.

## Figures and Tables

**Figure 1 pharmaceuticals-17-01008-f001:**
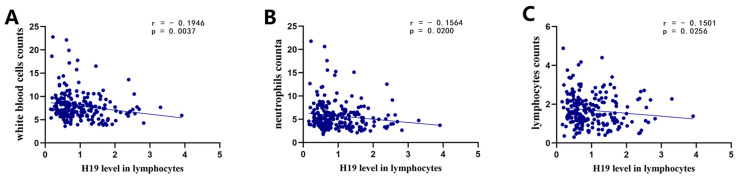
Correlation of H19 level in lymphocytes with counts of white blood cells, neutrophils, and lymphocytes. (**A**) Correlation between H19 level in lymphocytes and white blood cell counts; (**B**) correlation between H19 level in lymphocytes and neutrophils counts; (**C**) correlation between H19 level in lymphocytes and lymphocytes counts.

**Figure 2 pharmaceuticals-17-01008-f002:**
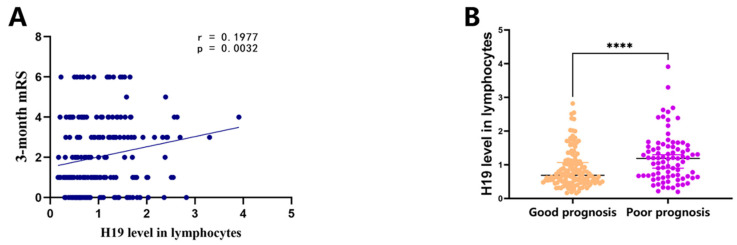
Correlation between H19 level in lymphocytes and prognosis in AIS patients. (**A**) Correlation between H19 level in lymphocytes and 3-month mRS scores. (**B**) Expression levels of H19 in lymphocytes of AIS patients with good prognosis (mRS 0–2) and with poor prognosis (mRS 3–6). AIS: acute ischemic stroke. mRS: modified Rankin scale. **** *p* < 0.0001.

**Figure 3 pharmaceuticals-17-01008-f003:**
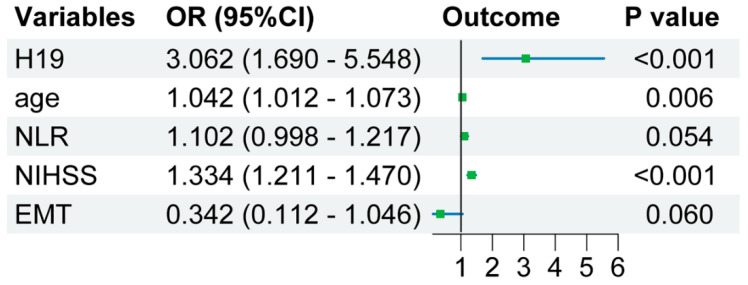
Forest plot for multivariate logistic regression analysis. OR—odds ratio; NLR—neutrophil-to-lymphocyte ratio; NIHSS—National Institutes of Health stroke scale; EMT—endovascular mechanical thrombectomy.

**Figure 4 pharmaceuticals-17-01008-f004:**
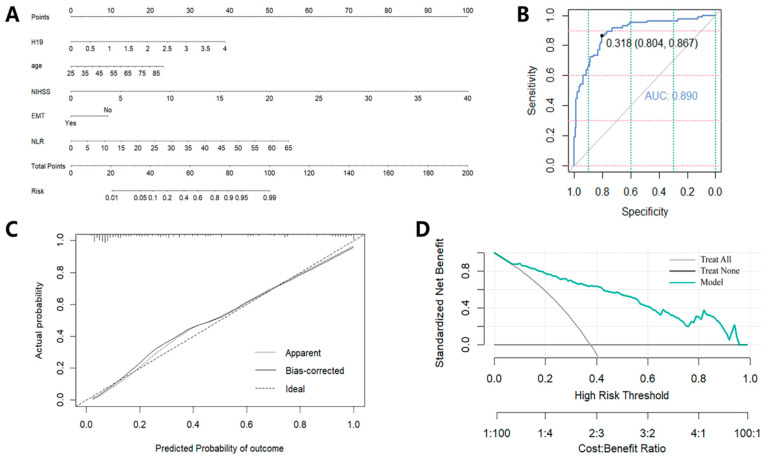
The creation and assessment of the nomogram. (**A**) The nomogram for predicting 3-month poor prognosis in AIS patients; (**B**) ROC curves of the nomogram; (**C**) calibration curves of the nomogram; (**D**) DCA analysis of the nomogram. EMT—endovascular mechanical thrombectomy; NLR—neutrophil-to-lymphocyte ratio.

**Table 1 pharmaceuticals-17-01008-t001:** Baseline characteristics of the whole study population according to the modified Rankin Scale (mRS) at 3 months (n = 221).

Baseline Characteristics	Total	Good Prognosis	Poor Prognosis	*p* Value
	(N = 221)	(N = 138)	(N = 83)	
Age, x ± s or median (IQR)	65.01 ± 13.17	62.50 [54.75–71.00]	70.00 [60.00–82.00]	<0.001 *
Male, n (%)	164 (74.21)	107 (77.54)	57 (68.67)	0.145
Onset-to-treatment time, h, median (IQR)	2.90 [1.45–5.05]	2.80 [1.40–4.70]	3.00 [1.70–6.30]	0.387
Systolic blood pressure, mm Hg, median (IQR)	150.00 [138.00–165.00]	150.00 [136.00–166.50]	148.50 [140.00–164.00]	0.703
Diastolic blood pressure, mm Hg, median (IQR)	83.00 [77.00–91.00]	83.00 [77.00–92.00]	86.00 [75.00–90.00]	0.536
Admission NIHSS scores, median (IQR)	6.00 [3.00–11.00]	4.00 [2.00–7.00]	12.00 [7.00–17.00]	<0.001 *
Treatment				
Intravenous thrombolysis, n (%)	93 (42.08)	64 (46.38)	29 (34.94)	0.095
EMT, n (%)	34 (15.38)	15 (10.87)	19 (22.89)	0.016 *
Prior risk factors				
Hypertension, n (%)	150 (67.87)	89 (64.49)	61 (73.49)	0.165
Diabetes mellitus, n (%)	75 (33.94)	41 (29.71)	34 (40.96)	0.087
Coronary heart disease, n (%)	48 (21.72)	21 (15.22)	27 (32.53)	0.003 *
Atrial fibrillation, n (%)	39 (17.73)	16 (11.59)	23 (27.71)	0.002 *
Stroke etiology				0.149
Large artery atherosclerosis, n (%)	117 (52.94)	77 (55.80)	40 (48.19)	
Small vessel occlusion, n (%)	55 (24.89)	37 (26.81)	18 (47.37)	
Cardioembolic, n (%)	10 (4.52)	4 (2.90)	6 (15.79)	
Other determined and undetermined, n (%)	39 (17.65)	20 (14.49)	19 (50.00)	
Clinical parameters				
White blood cells, ×10^9^/L, median (IQR)	7.38 [6.03–8.85]	7.21 [5.98–8.62]	7.87 [6.27–9.41]	0.104
Neutrophils, ×10^9^/L, median (IQR)	5.01 [3.84–6.52]	4.72 [3.70–5.93]	5.95 [4.04–7.56]	0.006 *
Lymphocytes, ×10^9^/L, median (IQR)	1.54 [1.12–2.16]	1.73 [1.25–2.22]	1.31 [0.91–1.90]	<0.001 *
NLR, median (IQR)	2.88 [2.03–5.61]	2.61 [1.77–4.33]	4.14 [2.34–7.43]	<0.001 *
Platelet counts, ×1000/mm^3^, median (IQR)	208.00 [169.00–244.50]	214.50 [171.00–250.75]	200.00 [163.00–236.00]	0.125
TG, mmol/L, median (IQR)	1.45 [0.96–2.53]	1.59 [1.03–2.70]	1.30 [0.84–2.01]	0.042 *
TC, mmol/L, median (IQR)	4.53 [3.72–5.41]	4.58 [3.85–5.48]	4.43 [3.60–5.10]	0.196
HDL, mmol/L, median (IQR)	1.18 (1.00–1.40]	1.20 [0.99–1.43]	1.17 [1.01–1.37]	0.524
LDL, mmol/L, x ± s or median (IQR)	2.69 [2.01–3.42]	2.78 ± 0.95	2.64 ± 0.93	0.282
Biomarkers				
H19 level in lymphocytes, median (IQR)	0.78 [0.55–1.33]	0.69 [0.51–1.07]	1.19 [0.66–1.54]	<0.001 *

IQR—interquartile range; NIHSS—national institutes of health stroke scale; EMT—endovascular mechanical thrombectomy; NLR—neutrophil to lymphocyte ratio; TG—triglyceride; TC—total cholesterol; HDL—high-density lipoprotein; LDL—low-density lipoprotein. * Statistically significant.

**Table 2 pharmaceuticals-17-01008-t002:** Univariate analyses of various potential prognostic factors in AIS patients.

	Univariate Analysis	
Parameter	OR (95% CI)	*p*
Age, y	1.045 [1.021–1.070]	<0.001 *
Male	1.547 [0.853–2.904]	0.146
Onset-to-treatment time, h	1.032 [0.964–1.105]	0.360
Systolic blood pressure, mm Hg	0.999 [0.988–1.010]	0.829
Diastolic blood pressure, mm Hg	0.997 [0.979–1.015]	0.766
Admission NIHSS scores	1.293 [1.203–1.389]	<0.001 *
Intravenous thrombolysis (%)	0.621 [0.354–1.089]	0.096
EMT (%)	0.411 [0.196–0.862]	0.019 *
Hypertension	1.527 [0.838–2.780]	0.167
Diabetes mellitus	1.642 [0.929–2.902]	0.088
Coronary heart disease	2.686 [1.398–5.162]	0.003 *
Atrial fibrillation	2.923 [1.439–5.939]	0.003 *
White blood cells, ×10^9^/L	1.064 [0.972–1.164]	0.176
Neutrophils, ×10^9^/L	1.119 [1.018–1.231]	0.020 *
Lymphocytes, ×10^9^/L	0.487 [0.324–0.732]	<0.001 *
NLR	1.176 [1.082–1.278]	<0.001 *
Platelet counts, ×1000/mm^3^	0.996 [0.991–1.001]	0.116
TG, mmol/L	1.049 [0.977–1.125]	0.187
TC, mmol/L	1.025 [0.933–1.127]	0.605
HDL, mmol/L	0.914 [0.398–2.101]	0.833
LDL, mmol/L	0.850 [0.633–1.142]	0.281
H19 level in lymphocytes	2.411 (1.510–3.849)	<0.001 *

OR—odds ratio; CI—confidence interval; NIHSS—National Institutes of Health stroke scale; EMT—endovascular mechanical thrombectomy; NLR—neutrophil to lymphocyte ratio; TG—triglycerideG; TC—total cholesterol; HDL—high-density lipoprotein; LDL—low-density lipoprotein. * Statistically significant.

**Table 3 pharmaceuticals-17-01008-t003:** Logistic regression analysis and additional predictive value of the model including H19 in lymphocytes for patients with poor prognosis (mRS > 3) at 3 months in the whole study population (n = 221).

	Clinical Model	Clinical Model + H19
Logistic regression		
R^2^ (Cox and snell)	0.366	0.409
Age, y	OR = 1.040 [1.011–1.069], *p* = 0.006 *	OR = 1.042 [1.012–1.073], *p* = 0.006 *
Admission NIHSS scores	OR = 1.330 [1.212–1.459], *p* < 0.001 *	OR = 1.334 [1.211–1.470], *p* < 0.001 *
EMT	OR = 3.339 [1.123–9.930], *p* = 0.030 *	OR = 0.342 [0.112–1.046], *p* = 0.060
NLR	OR = 1.096 [0.998–1.203], *p* = 0.055	OR = 1.102 [0.998–1.217], *p* = 0.054
H19 level in lymphocytes	-	OR = 3.062 [1.69–5.548], *p* < 0.001
AUC		
	0.870	0.890
IDI index, %		
Total IDI	-	5.03 [1.76–8.30]
*p* value	-	0.0026 *
NRI index, %		
Continuous NRI	-	68.33 [42.86–93.79]
*p* value	-	<0.0001 *

OR—odds ratio; mRS modified Rankin scale; NIHSS—National Institutes of Health stroke scale; EMT—endovascular mechanical thrombectomy; NLR—neutrophil to lymphocyte ratio; AUC—area under the curve; IDI—integrated discrimination improvement; NRI—net reclassification improvement. * Statistically significant.

## Data Availability

The data and materials that support the findings of this study are available from the corresponding authors upon reasonable request. The data are not publicly available due to privacy or ethical restrictions.

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
