# Peer review of "Long Non-Coding H19 in Lymphocytes: Prognostic Value in Acute Ischemic Stroke Patients"

_pharmaceuticals, 2024, doi:10.3390/ph17081008_

Round 1
Reviewer 1 Report
Comments and Suggestions for Authors
This manuscript by Zhong et al. describes the association of lncRNA H19 expression in lymphocytes at the onset of acute ischemic stroke with unfavorable progression after three months based on mRS scores. This fact could have prognostic value and provide clues about the therapeutic strategies to use with these patients.
This finding is in line with that described by others in plasma and exosomes (see Wang J, Cao B, Gao Y, Chen YH, Feng J. Exosome-transported lncRNA H19 regulates insulin-like growth factor-1 via the H19/let-7a/insulin-like growth factor-1 receptor axis in ischemic stroke. Neural Regen Res. 2023 Jun;18(6):1316-1320. doi: 10.4103/1673-5374.357901. PMID: 36453417; PMCID: PMC9838162), but the authors analyze the expression of H19 specifically in lymphocytes. Other studies have also shown an association of H19 with similar clinical situations (see Lapikova-Bryhinska T, Ministrini S, Puspitasari YM, Kraler S, Mohamed SA, Costantino S, Paneni F, Khetsuriani M, Bengs S, Liberale L, Montecucco F, Krampla W, Riederer P, Hinterberger M, Fischer P, Lüscher TF, Grünblatt E, Akhmedov A, Camici GG. Long non-coding RNAs H19 and NKILA are associated with the risk of death and lacunar stroke in the elderly population. Eur J Intern Med. 2024 May;123:94-101. doi: 10.1016/j.ejim.2023.11.013. Epub 2023 Nov 18. PMID: 37981527). Anyway, the study by Zhong et al. is valuable because of the methodology used to demonstrate the independent association of H19 with the severity of sequelae after ischemic stroke.
I would like to make some observations and questions to the authors that may bring more clarity to their data and the conclusions they draw from it.
In the Methods Section, sub-section 2.4. cDNA synthesis and reverse transcription-polymerase chain reaction (RT-PCR) it is described the procedure to analyse H19 by quantitative PCR. According to the provided information, it seems that a SYBR green protocol has been used, but it is not made explicit. Moreover, the reagents (kit) used to do the amplifications should be indicated as well as if technical replicates were performed for each sample. The reference gene expression chosen to normalize the results is beta-actin, but what reference condition was used to obtain the relative expression calculated using the ddCt method?
In the Results section, sub-section 3.1. Baseline characteristics of patients grouped by 3-month functional outcomes, the data of some variables under comparison in groups are expressed in a non-homogenous way. Age, lymphocyte count, platelet count, total cholesterol concentration, HDL cholesterol concentration (Table 1), are expressed as mean+/-SD or median[IQR] depending on the group. This shows that the distribution of the variables does not conform to normality in one of the compared groups, but it may be confusing for the reader. I suggest choosing only one way to express these results, which could be the median [IQR] if non-normality happens.
A reference to a Table 2 is made in sub-section 3.2. Relationship between the H19 expression level and clinical manifestations, but I have not found it in the manuscript.
The same happens for a Table 3, which is cited in sub-section 3.3. Incremental predictive value of lncRNA H19 in lymphoctes for poor prognosis of AIS.
The Discussion section lists the analyses and results obtained, but I miss some arguments about the meaning and significance of the association of increased H19 expression specifically in lymphocytes and a poor outcome for AIS. Could the authors provide some biological reasoning for this? Additionally, they could discuss the potential advantages of determining H19 in lymphocytes compared to plasma/serum or extracellular vesicles. Do they have data of H19 quantitation in plasma/serum of the patients to study correlation?
Comments on the Quality of English Language
The usage of English language is correct, only minor editing is required.
Author Response
Comments 1: In the Methods Section, sub-section 2.4. cDNA synthesis and reverse transcription-polymerase chain reaction (RT-PCR) it is described the procedure to analyse H19 by quantitative PCR. According to the provided information, it seems that a SYBR green protocol has been used, but it is not made explicit. Moreover, the reagents (kit) used to do the amplifications should be indicated as well as if technical replicates were performed for each sample. The reference gene expression chosen to normalize the results is beta-actin, but what reference condition was used to obtain the relative expression calculated using the ddCt method?
Response 1: Thanks for your suggestions. We have elaborated on these in the revised manuscript, Please see “2.4. cDNA synthesis and real-time polymerase chain reaction (RT-PCR)” of the revised manuscript.)
Comments 2:In the Results section, sub-section 3.1. Baseline characteristics of patients grouped by 3-month functional outcomes, the data of some variables under comparison in groups are expressed in a non-homogenous way. Age, lymphocyte count, platelet count, total cholesterol concentration, HDL cholesterol concentration (Table 1), are expressed as mean+/-SD or median[IQR] depending on the group. This shows that the distribution of the variables does not conform to normality in one of the compared groups, but it may be confusing for the reader. I suggest choosing only one way to express these results, which could be the median [IQR] if non-normality happens.
Response 2: Thank you for your highly professional advice. We strongly agree with your perspective. And we have corrected this mistake in the revised manuscript. Two groups of consecutive data, if one of the groups is non-normally distributed, then both groups of data are expressed as mean+/-SD or median[IQR]. We have revised the manuscript, please see “3.1 Baseline characteristics of patients grouped by 3‐month functional outcomes.” of the revised manuscript.
Comments 3: A reference to a Table 2 is made in sub-section 3.2. Relationship between the H19 expression level and clinical manifestations, but I have not found it in the manuscript.The same happens for a Table 3, which is cited in sub-section 3.3. Incremental predictive value of lncRNA H19 in lymphoctes for poor prognosis of AIS.
Response 3: Thank you for your valuable comments. We have added Table 2 and Table 3 in the revised manuscript. Please see the revised manuscript and revised tables. We sincerely apologize for any confusion or suboptimal reading experience this may have caused.
Comments 4:The Discussion section lists the analyses and results obtained, but I miss some arguments about the meaning and significance of the association of increased H19 expression specifically in lymphocytes and a poor outcome for AIS. Could the authors provide some biological reasoning for this? Additionally, they could discuss the potential advantages of determining H19 in lymphocytes compared to plasma/serum or extracellular vesicles. Do they have data of H19 quantitation in plasma/serum of the patients to study correlation?
Response 4: Thanks for your constructive suggestions. We have added the arguments about the meaning and significance of the association of increased H19 expression specifically in lymphocytes and a poor prognosis for AIS. In addition, we have also added the potential advantages of determining H19 in lymphocytes compared to plasma/serum or extracellular vesicles. Previous studies have revealed the prognostic value of H19 in plasma and exosomes in AIS (PMID:37981527, PMID: 36453417). Hence, we choose to focus on the prognostic value of H19 within lymphocytes in acute ischemic stroke patients rather than H19 in plasma/serum. Thanks for your meaningful comments again, and also thanks for your thoughtful considering our study. We have revised as you suggested, please see “Introduction” and “Discussion” of the revised manuscript..
Reviewer 2 Report
Comments and Suggestions for Authors
The article "Long non-coding H19 in Lymphocytes: Prognostic Value in Acute Ischemic Stroke Patients" presents a promising exploration of lncRNA H19 as a prognostic biomarker for acute ischemic stroke (AIS) patients. The study employs acceptable methodologies, such as qRT-PCR and logistic regression analysis, to establish the prognostic significance of lncRNA H19. The findings, including the negative correlation with white blood cell counts and positive correlation with mRS scores, are intriguing and suggest potential clinical applications. However, the manuscript has several shortcomings that warrant revisions. The presentation of methods and findings could be more detailed to enhance reproducibility and clarity. Additionally, the manuscript contains several typographical errors that need correction. Addressing these issues will improve the overall quality and reliability of the study, making it more suitable for publication.
1. In the inclusion criteria, under section 2.1, it is stated that diagnostic imaging results from both CT and MRI were considered. Please elaborate on this statement. Were CT and MR imaging results used simultaneously as inclusion criteria? Or does the manuscript intend to say that either computed tomography OR magnetic resonance imaging findings were used for confirmation of diagnosis?
2. Certain sections of the methods are written objectively and warrant revision. For instance, in section 2.3, the finishing lines of the first paragraph are written objectively and use modal verbs (should), which is considered improper. Please explain your methodology as it were carried out, not as it should have been carried out.
3. In Table 1, the two groups being compared to each other are labeled as "favorable outcome" and "unfavorable outcome". However, throughout the text, the two groups are referred to as "goo prognosis" and "poor prognosis". Please ensure that the groups being studied are mentioned consistently in the text to avoid potential confusion.
4. In section 3.2, the results of multivariate logistic regression analysis for 5 variables in question are only mentioned in the text, without any visualizations. While the manuscript does provide a nomogram in Fig. 3A, this should not exempt the manuscript from providing relevant graphs for backward stepwise regression method.
5. Descriptions provided in section 3.4. regarding the methodological approach to illustrating a nomogram are redundant and should be removed. If necessary, these details should be provided in the methods section.
6. Several typographical errors can be spotted throughout the text. The abstract includes an incomplete sentence ("while a positive correlation with the mRS scores three months poststroke via spearman correlation analysis"). Additionally, authors are discouraged from overusing certain words and phrases such as "meticulous".
Comments on the Quality of English LanguageOverall, the quality of the written language is fair. However, certain typographical errors are present throughout the paper that warrant revision.
Author Response
Comments 1: In the inclusion criteria, under section 2.1, it is stated that diagnostic imaging results from both CT and MRI were considered. Please elaborate on this statement. Were CT and MR imaging results used simultaneously as inclusion criteria? Or does the manuscript intend to say that either computed tomography OR magnetic resonance imaging findings were used for confirmation of diagnosis?
Response 1: Thank you for your instructions. We are grateful for your meticulous review of our manuscript and for pointing out the errors within it. We have revised this text in the “2.1. Study participants” (please see the revised manuscript).
Comments 2: Certain sections of the methods are written objectively and warrant revision. For instance, in section 2.3, the finishing lines of the first paragraph are written objectively and use modal verbs (should), which is considered improper. Please explain your methodology as it were carried out, not as it should have been carried out.
Response 2: Thanks for your valuable suggestion. We have rephrased the text accordingly, please see the revised manuscript.
Comments 3: In Table 1, the two groups being compared to each other are labeled as "favorable outcome" and "unfavorable outcome". However, throughout the text, the two groups are referred to as "goo prognosis" and "poor prognosis". Please ensure that the groups being studied are mentioned consistently in the text to avoid potential confusion.
Response 3:Thank you for your valuable comments. We have revised our manuscript according to your suggestion, please see the revised manuscript
Comments 4: In section 3.2, the results of multivariate logistic regression analysis for 5 variables in question are only mentioned in the text, without any visualizations. While the manuscript does provide a nomogram in Fig. 3A, this should not exempt the manuscript from providing relevant graphs for backward stepwise regression method.
Response 4:Thanks for your reminding and beg your pardon. We have added Table 2 and Table 3 in the revised manuscript. Please see the revised manuscript and revised tables. We sincerely apologize for any confusion or suboptimal reading experience this may have caused.
Comments 5: Descriptions provided in section 3.4. regarding the methodological approach to illustrating a nomogram are redundant and should be removed. If necessary, these details should be provided in the methods section.
Response 5:Thank you for your valuable comments. We have accordingly deleted these, please check the revised manuscript for details.
Comments 6: Several typographical errors can be spotted throughout the text. The abstract includes an incomplete sentence ("while a positive correlation with the mRS scores three months poststroke via spearman correlation analysis"). Additionally, authors are discouraged from overusing certain words and phrases such as "meticulous".
Response 6:Thanks for your constructive comments. We have checked the language and spelling mistakes seriously, and we have improved language of the revised manuscript carefully. Thank you for your meticulous review and comments again.
Round 2
Reviewer 1 Report
Comments and Suggestions for Authors
The authors have adequately addressed the questions and suggestions I made in my review report. I believe that the current version of the manuscript has been improved and deserves publication in Pharmaceuticals.
Reviewer 2 Report
Comments and Suggestions for Authors
I would like to thank the authors for addressing my comments. I have no further comments.